# Uncertainty Analysis of an Optoelectronic Strain Measurement System for Flywheel Rotors

**DOI:** 10.3390/s21248393

**Published:** 2021-12-16

**Authors:** Matthias Franz Rath, Bernhard Schweighofer, Hannes Wegleiter

**Affiliations:** Energy Aware Measurement Systems Group, Institute of Electrical Measurement and Sensor Systems, Graz University of Technology, 8010 Graz, Austria; bernhard.schweighofer@tugraz.at (B.S.); wegleiter@tugraz.at (H.W.)

**Keywords:** strain measurement, flywheel, CFRP

## Abstract

The strain in a fast spinning carbon fiber flywheel rotor is of great interest for condition monitoring, as well as for studying long-term aging effects in the carbon fiber matrix. Optoelectronic strain measurement is a contactless measurement principle where a special reflective pattern is applied to the rotor which is scanned by a stationary optical setup. It does not require any active electronic components on the rotor and is suited for operation in a vacuum. In this paper, the influences of the key parts comprising the optoelectronic strain measurement are analyzed. The influence of each part on the measurement result including the uncertainty is modeled. The total uncertainty, as well as each part’s contribution is calculated. This provides a valuable assessment of requirements for component selection, as well as tolerances of mechanical parts and processes to reach a final target measurement uncertainty or to estimate the uncertainty of a given setup. We have shown that the edge quality of the special reflective pattern has the strongest influence, and how to improve it. Considering all influences, it is possible to measure strain with an uncertainty of less than 1% at a rotation speed of 500Hz.

## 1. Introduction

A flywheel energy storage system (FESS) is an electro-mechanical system for short-term energy storage. It consists of a rotating mass (flywheel) coupled to an electric motor/generator. Rotational energy is stored in the flywheel by accelerating it with the motor. Energy is reclaimed by reconfiguring the motor as a generator and decelerating the flywheel.

The energy capacity of a FESS is determined by the moment of inertia of the flywheel and its speed. To achieve higher speeds, flywheels have to be manufactured out of high-strength materials such as carbon-fiber-reinforced plastic (CFRP) to withstand the rotational strain. Continuous health monitoring of the CFRP flywheel is required to ensure safe operation. The flywheel we are developing consists of a cylindrical inner steel core with press-fitted outer CFRP rings. Contactless measurement methods are required to avoid influencing the rotational behavior of the flywheel and to avoid additional points of friction. We also want to avoid mechanical weakening of the flywheel axle or CFRP rings and therefore adding sensors to the rings is not an option.

Established methods for non-destructive testing of CFRP flywheels such as ultrasonic, X-ray, thermography, and visual testing focus on detecting manufacturing flaws in materials [1,2,3]. Ultrasonic testing requires a well-coupled contact to the device under test, X-ray requires specialized equipment and safety considerations. Thermography and visual testing are limited to low speeds by the frame rate of the cameras.

Additionally, methods for strain measurement are also applicable for health monitoring. Strain gauges can be applied to the rotor surface [4,5,6] or between layers of carbon fibers. Since data transmission by slip rings is not practical at high speeds because of self-heating by friction [7], a telemetry system is required to record and transmit the data. The electronics of a telemetry system have to be placed close to or even inside the axle to reduce the rotational forces on the electronics. This has the drawback of weakening the flywheel axle or its CFRP rings.

Electronic speckle pattern interferometry can be used to measure stress contactlessly. However, it relies on cameras which are impractical at high rotation speeds because of their frames-per-second limit [8]. Laser Doppler velocimetry uses photo diodes instead of cameras, which would solve the speed issue but relies on positioning on a suitable speckle [9].

Optoelectronic strain measurement (OESM) relies on a special printed pattern on the flywheel which is read remotely by an optoelectronic system [10]. There are no electronics required on the flywheel itself but only on the stationary housing. On the downside, the measurement principle relies on correct and robust paint application [11].

The optoelectronic strain measurement method suits all of our requirements and is further investigated in this paper. There are many sources and parameters that can influence the OESM principle and the uncertainty of the final measurement. In this paper, we try to identify and quantify common sources of measurement uncertainty.

Our research is conducted by the following methods: The optical part of the OESM is simulated via ray tracing and the results are cross-checked via an experimental measurement setup. Sources of measurement uncertainty are identified and modeled. Then, a Monte Carlo simulation approach is used to evaluate each contribution to the final measurement uncertainty.

The paper is organized as follows: Section 2 gives an overview of the working principle of OESM and deals with the required measurement accuracy. Section 3 describes the experimental optical setup for reflectivity measurement. Section 4 describes the simulation methods used to calculate the uncertainties of the final measurement while Section 5 shows the results for each influence source. Finally, in Section 6, the achieved results are discussed and the 1% strain measurement uncertainty is shown.

## 2. State of the Art of Optoelectronic Strain Measurement

This section explains the principle operation of an optoelectronic strain measurement (OESM) method for flywheel rotors as shown in Figure 1 [10]. A pattern of spiral sections is applied to the top surface of the rotor. This pattern is locally illuminated in a small spot by a stationary light source and the reflected light is measured by a sensor. The intensity of the reflected light depends on the surface it is reflected from. The dark CFRP surface reflects less light than the chrome paint used to apply the pattern.

Over each full rotation of the flywheel, a characteristic sequence of light and dark pulses is measured along the circumference of the pattern, as illustrated in Figure 2. Strain in the rotor, for instance from centrifugal forces, causes the rotor and also the applied pattern to deform and stretch radially. Due to this deformation, the stationary light source and sensor now see another part of the pattern, and the relation of light and dark pulses in the signal changes. From the duty cycle of the light pulses, the position on the pattern and therefore the relative deformation of the flywheel can be calculated.

The special pattern, shown in Figure 1, consists of lobes: filled, three-edged shapes where two edges are pieces of Archimedean spirals [12]. In polar coordinates, an Archimedean spiral is described by its constant relation of the radius *r* and the angle θ. Lobes are repeated over the circumference and radius, providing multiple measurement positions on the flywheel.

The duty cycle *D* is
(1)$$D=\frac{t_{chrome}}{t_{lobe}}=\frac{t_{chrome}}{t_{chrome}+t_{CFRP}}$$
where $t_{chrome}$ is the measured width of the light pulse, $t_{CFRP}$ the width of the dark pulse and $t_{lobe}$ is the width of one lobe.

To measure the deformation *u* at a sensor position, we have to determine its radial position within a lobe from the respective duty cycle measurement *D*. The radial position on a lobe segment *r* can be calculated as
(2)$$r=r_{o}-D\;\cdot \;\left(r_{o}-r_{i}\right)$$
where $r_{o}$ and $r_{i}$ are the outer and inner radii of the lobe segment under the sensor. Reference values for each sensor position are acquired at low speeds where the deformation is near zero. The local deformation *u* at each sensor position can then be determined by the difference of the reference value and the instantaneous measurement.

Systematic influences of the flywheel rotor such as in-plane and out-of-plane vibrations can be corrected by using multiple sensors at different angular sensor positions and utilizing calibration patterns on the rotor surface [10]. An expansion to include a more detailed rotor behavior is possible but would exceed the scope of this paper [13]. The following common influence sources collected from literature and preliminary works are investigated:Paint edge characteristics;Chrome paint and CFRP reflectivity;Illumination spot size;Photodetector angle;Illumination source power;Edge transition threshold detection.

### 2.1. Accuracy Requirements

In this section, the requirements for the OESM and the total uncertainty of the detected threshold position are discussed. We use the flywheel design from our research project Flygrid as reference for size and composition to define the boundary conditions for the OESM system [14]. This flywheel consists of an inner steel core with a radius of 120 mm with three press-fitted CFRP rings, as illustrated in Figure 3 [4,15,16,17,18]. The final outer radius is 260 mm. The CFRP rings’ relative deformation is dependent on the radius. Calculation of the radial stress profile in the CFRP rings is done according to [19] and plotted in Figure 4. The calculations show a maximum deformation of 1029 μm at the nominal rotation speed of 500 Hz.

We are especially interested in the radial strain because the CFRP rings are press-fitted into each other and could become loose when the properties of the CFRP change, for example by aging effects [20,21]. The radial strain $\varepsilon_{r}$ at the fixed radial location $r^{\prime }$ is calculated as
(3)$$\varepsilon_{r}={\left.\frac{du}{dr}\right|}_{r=r^{\prime }}$$
where du is the relative deformation between two measurement locations and dr is the radial distance between two measurement locations.

Since the relative deformation becomes smaller for the outer rings, the worst case for radial strain measurement is in the outer ring. At the interface between the middle and outer ring, the relative deformation is du=121μm for a distance dr=20 mm between two sensors. The distance dr was chosen as small as possible for the best spatial resolution while still allowing enough space for mounting hardware between OESM sensor systems whose size is mostly determined by the diameter of the laser diode (8 mm). We chose a maximum allowed uncertainty of 1% for the strain measurement at the interface between the middle and outer ring. This choice provides a good compromise between a high resolution and the necessary complexity of the measurement system. Parameters of the OESM pattern design, such as the steepness and the number of the pattern lobes (repetitions over the circumference), influence the sensitivity. The steepness is influenced by the radial dimensions of the pattern (repetitions over the radius). In this paper, we use four lobes repeated over the circumference, so we obtain one strain measurement per quarter section of the flywheel. With a distance dr=20 mm between sensors, we can repeat the pattern seven times over the radius.

In this paper, we focus on the uncertainty with which the threshold position of the transition between CFRP and the painted pattern surface can be measured. The threshold position has to be measured with a resolution of
(4)$$d_{\Delta ,Thresh}=\frac{2\pi }{N_{lobe}}\;\cdot \;\frac{1}{dr}\;\cdot \;u_{min}\;\cdot \;r=19.8\mu m$$
where $N_{lobe}$ is the number of lobes in the pattern, $u_{min}=1.2\mu m$ is the minimum deformation we want to measure (1% of du at the interface between the middle and outer CFRP ring) and r=210 mm is the radial position of the interface between the middle and outer CFRP ring [10].

To achieve a high confidence in our measurement, we chose a k-factor of 3 for the maximum allowable uncertainty. The ±3·σ interval of the measurement uncertainty has to be less than the required resolution $d_{\Delta ,Thresh}$. For a normal distribution, 99.73% of the measurements then are within $\pm d_{\Delta ,Thresh}/2$ of the actual position. The maximum uncertainty in threshold position allowed to achieve the required 1% accuracy in radial strain measurement is therefore
(5)$$\sigma_{\Delta ,Thresh,max}=\frac{d_{\Delta ,Thresh}}{2\;\cdot \;3}=3.3\mu m$$

## 3. Measurement Setup

The reflective properties of the CFRP flywheel surface and the reflective paint are important features of the OESM system and are also required parameters to set up the simulation model. A measurement setup was created to characterize the behavior of these surfaces under known conditions similar to those of an OESM system. Because the directional reflectivity is of great interest for the uncertainty analysis of the OESM system, the measurement setup has to allow for different incidence angles as well as different measurement angles.

The picture in Figure 5 shows the measurement setup and Figure 6 illustrates the involved angles and distances. All elements are arranged on a rail to keep them aligned. The illumination source is a laser in the visible red spectrum (650 nm) with integrated focusing lens (laser pointer). A visible wavelength was chosen for easy alignment and focusing in an experimental setup. The laser is operated with a constant current of 7 mA which was chosen so that the photo diode is not saturated when directly hit by the laser.

The laser is focused on the surface of the sample under test. The sample holder can be rotated to different incidence angles. The sensor assembly can carry different sensors via 3D printed adapter pieces. It can be rotated 360° and has a scale marking every 2.5° with an estimated positioning error due to the scale of less than 0.5°. The sensor distance can be varied but is limited by the size of the sensor which could collide with the sample at certain angles.

A square photo diode was used as light detector for this work. It has an active sensing surface of 3 mm by 3 mm, and its sensitivity peak at 650 nm matches the wavelength of the laser. The photo diode is operated in reverse bias and the light-dependent reverse diode current $I_{PD}$ is fed into an oscilloscope input (4444 USB oscilloscope by Pico Technology), shown in Figure 7. The resulting voltage drop on the input resistor $R_{input}=10\;M\Omega$ is measured. The voltage $V_{Bat}=9\;\mathrm{volt}$ is supplied by a battery to avoid line-bound interferences.

For reference measurements, the sample holder is removed and the laser is aimed directly at the photo diode. The optical path length is kept constant. Reference measurements are required to calculate the directional reflectivity as a ratio between reflected light measurement and direct sensor illumination measurement. Direct illumination was also used to acquire long-term drift measurements of the laser and photo diode.

## 4. Simulation Framework

This section describes the simulation techniques used to investigate the uncertainty of the strain measurement from different sources. The core of the investigation is a simulation of the elements of the OESM system—light source, reflective surface, and photo detector. We want to model the light source as a laser beam which can be approximated by a Gaussian beam. Different reflective surfaces can be modeled by how they influence the reflected beam. Although Gaussian beam propagation can be calculated analytically, available frameworks deal with discrete optical components, such as lenses or mirrors. For our investigations, we have to simulate reflections of the Gaussian beam on surfaces whose characteristics are inhomogeneous and location-dependent. We decided on a ray tracing approach where the characteristics of the Gaussian beam are approximated via the distribution and direction of individual rays.

The steps we did for each source of uncertainty are outlined in Figure 8. The standard deviations of the detected threshold position (common metric) are calculated for each source of uncertainty and represent the influences on the final measurement result. The allowable limits of uncertainty were discussed in Section 2.1.

### 4.1. Ray Tracing Simulation

To model the reflection of a laser beam on an interface edge between two materials with different reflective properties, a 3D ray tracing simulation was implemented in Matlab. The OESM system consists of only three relevant components: the illumination source, the reflective surface, and the receiver. These components are arranged as shown in Figure 6. The interaction between the light rays and the reflective surface is modeled by implementing the laws of reflection.

A laser beam can be approximated by a Gaussian beam which models the widening of the beam, as well as the intensity distribution within the beam. Since we are interested in the interaction of a Gaussian beam with the transition region between CFRP and chrome paint, the Gaussian beam is discretized into rays. The characteristics of a Gaussian beam can be approximated by ray distribution (intensity) and randomization of ray directions (beam widening) [22].

Reflection of the rays on the material surface, such as CFRP and chrome paint, is modeled by a combination of directional diffuse reflection and Lambertian diffuse reflection. For diffuse reflections, ray tracing was not considered practical because most of the diffusely reflected rays would never reach the detector. The higher number of rays required to produce results without quantization artifacts would lead to longer simulation times. Diffuse reflection was implemented by averaging the power of the incoming rays on the reflective surface, weighted by the respective diffuse reflection coefficients of the surface materials. This average power was then scaled by the fraction of area the sensor surface occupies in the half sphere above the reflective surface. Directional diffuse reflection is modeled through widening of the reflected beam.

Figure 9 and Figure 10 show comparisons of the measurement results and the simulation results for the reflection coefficient $R_{0}$ over different emergent angles α for chrome paint as well as CFRP.

### 4.2. Monte Carlo Method

After modeling each of the influence parameters affecting the OESM system within the ray tracing framework, their influence on the measurement uncertainty of the threshold position measurement has to be investigated. A Monte Carlo approach was chosen because it allowed us to use a simple, straightforward ray tracing model. The ray tracing simulation is repeated 1000 times for each influence parameter For each repetition, a new random parameter value is generated from its distribution. This results in a set of possible transition functions for each influence parameter whose statistics are analyzed in turn. The value of interest is the deviation of the threshold position. Each simulated transition function is scaled to a percentile representation where 0% is the average minimum value and 100% corresponds to the average maximum value. The threshold is set at 50% and the position at which the threshold is reached is calculated for each repeated simulation.

An exemplary distribution of the threshold positions for 1000 simulation repetitions is shown in Figure 11. The parameter that was investigated in this example is the random roughness of the paint edge. The paint edge roughness has the standard deviation $\sigma_{edge,A}=27.3\mu m$, and the distribution and frequency spectrum are matched with those of our measurements. From the distribution of resulting threshold positions, the standard deviation $\sigma_{\Delta ,thresh}=18.9\mu m$ is calculated.

Because of the discretization of the ray tracing simulation and the randomized ray generation, there is noise in the simulation itself. This simulation noise can be determined by running the Monte Carlo simulation with all influence parameters set to constant values. The distribution of threshold positions caused by the ray tracing approach, resulting in uncertainty due to simulation noise $\sigma_{\Delta ,thresh}=0.6\mu m$, is shown in Figure 12. For an uncertainty influence simulation to be considered valid, its resulting uncertainty has to be significantly larger than the simulation noise.

## 5. Uncertainty Analysis

This section aims to describe common sources of uncertainty in the strain measurement result. For each source, a model is implemented within the simulation framework. The resulting uncertainty of the threshold position is then calculated by following the steps described in Section 4.

### 5.1. Paint Edge Characteristics

The paint edges are the transitions between the CFRP and the chrome paint when viewed from the top of the flywheel. These paint edges are not perfectly straight at microscopic levels but show a rough edge, as illustrated in Figure 13. In the simulation, we model the edge roughness as a random variable with 0 mean, which is added to the ideal edge line. There are two parameters for the edge roughness random variable: Its amplitude distribution and its spectral distribution. To produce realistic model parameters for paint edges, we want to determine those parameters from actual paint samples.

Two paint sample variants were photographed under a USB microscope. In Variant A, a section of a CFRP sample was masked off with electrical tape, then spray-painted with chrome paint, and finally the masking tape was removed while the paint was still wet. This method was chosen because a preliminary test has shown that removing the tape after the paint has dried leads to excessive paint pooling at the tape edge, as well as chipping on the paint edge. For Variant B, the same procedure as in Variant A was used, but the paint edge was reworked by scratching the edge with a blade along a straight edge.

When comparing the microscope photo of Variant A in Figure 14 with that of Variant B in Figure 15, the edge in Variant B looks straighter and has less paint chipping. The histogram analysis of the edges’ amplitude distributions are in good agreement with a normal distribution. The amplitude standard deviation for Variant A is $\sigma_{edge,A}=27.3\mu m$ and for Variant B $\sigma_{edge,B}=12.6\mu m$.

A spectral analysis of the measured edge Variant A is shown in Figure 16. The amplitudes of the spectral components become smaller with rising frequency. To model the behavior of the paint edge in the simulation framework, random signals with the same power spectral density as the measured spectrum are generated and scaled to the desired standard deviation. An example of such a generated signal is shown in Figure 17.

The results of a parameter variation are shown in Figure 18. The range of $\sigma_{edge}$ values were deliberately chosen to illustrate the effect on the paint edge transition function which becomes less steep for larger values of $\sigma_{edge}$. Simulations with the realistic values for Variants A and B are illustrated in Figure 19. The mean value over all Monte Carlo simulation runs lies on the straight paint edge, but the uncertainty increases for larger values of $\sigma_{edge}$.

### 5.2. Chrome Paint and CFRP Reflectivity

The reflectivity of the different surfaces is not a constant value but can change with position. This is due to inhomogeneities in the CFRP or chrome paint structure. We model the variation as a multiplicative random variable that scales the average reflectivity $R_{0}$. The surface upon which the incoming rays are reflected is divided into a grid. Each point on the grid is assigned a random scaling value, following a distribution derived from measurements.

Measurement data from a previous paper was analyzed to attain realistic values for the random reflectivity model parameters [11]. In this previous experiment, the transition function over a paint edge was measured with a laser and a photo diode. The CFRP sample had a section of chrome paint applied which was moved under the laser/sensor assembly by a stepper motor in 10μm increments, and the signal of the photo diode sensor was recorded. Measurement results are shown in Figure 20. The standard deviation of the multiplicative reflectivity random variable $\sigma_{R0}=12.7\%$ for chrome paint was calculated.

Spectral analysis shown in Figure 21 indicates a low-pass behavior. To model the deviation in reflectivity $R_{0}$ in the simulation studies, a two-dimensional grid of white Gaussian noise is generated and subsequently low-pass filtered. A moving average filter of length 5 is applied in both dimensions, one after the other. The generated random signal is then scaled to match the standard deviation of the measured chrome sample. An example of the generated random signal, in comparison to the measured behavior of $R_{0}$, is shown in Figure 20.

The influences of different standard deviation values for the randomized $R_{0}$ are illustrated in Figure 22 where the range of values is deliberately chosen to demonstrate the effect. Simulation results for a realistic range of values, as derived from the measurement of a chrome sample, are shown in Figure 23.

### 5.3. Illumination Spot Size

This section investigates the influence of the illumination spot size. The illumination spot refers to the area illuminated by the light source on the flywheel surface. It is of special interest because the noisy properties of the reflective surface are averaged over the illumination spot.

Although the shape of the laser spot is generally elliptic, our preliminary examinations have shown a nearly circular spot which is subsequently assumed for the simulation study [11]. The illumination spot size is then only dependent on the distance *h* as shown in Figure 6. For the scope of this paper, we assume the beam diameter and the opening angle of the laser are constant. The distance *h* can change by $\Delta_{h}$ because of tolerances in the flywheel manufacturing process, tilting of the elastically mounted flywheel at certain speeds, or deformation of the flywheel.

The influence of $\Delta_{h}$ is demonstrated in Figure 24. With larger $\Delta_{h}$, the illumination spot also becomes larger. $\Delta_{h}=200\mu m$ is the specified height tolerance for the CFRP part of the flywheel rotor. The shift in $\Delta_{thresh}$ caused by this potential change in *h* is substantial but constant, and can be reduced by calibration. Calibration is possible by additional measurement of the flywheel’s radial and axial position and by including reference marks of constant width on the OESM pattern [10]. The inductive distance sensors we use for axial position measurements show a measurement noise of 3.4 μm when averaging over a full flywheel revolution at the maximum speed of 500 Hz. Because of the 100 kg mass of the flywheel, we can assume that *h* only changes slowly compared to the rotation speed, and can, therefore, be assumed constant during one revolution. This leaves the measurement noise of the inductive distance sensor as the uncertainty $\sigma_{\Delta h}$ of the height measurement. The influence of a random variation of $\Delta_{h}$ was investigated and is illustrated in Figure 25.

### 5.4. Photodetector Angle

In this section, the mounting angle of the sensor is investigated. We are interested in maximizing the sensor power swing between the maximum and the minimum of the edge transition function for the best separation between CFRP and chrome paint. The results of the simulation for different deviations $\Delta_{\alpha }$ is shown in Figure 26. The largest separation is achieved for $\Delta_{\alpha }=-12{}^{\circ }$. This is because of the directional reflectivities of chrome paint and CFRP, shown in Figure 9 and Figure 10, which are Gaussian functions with added offsets. It can be concluded that for an incidence angle ϕ= 45°, the best sensor position is not at an emerging angle α= 45° but rather at a position shifted by $\Delta \alpha =-12{}^{\circ }$.

Random variation of the sensor angle results in additional uncertainty in the measurement of the threshold position $\Delta_{thresh}$. With rigid mounting of the sensor assembly, the variation of the sensor angle due to vibrations is unlikely. During sensor assembly we have to ensure that the uncertainty of the angle $\sigma_{\Delta \alpha }$ is less than $0.2{}^{\circ }$ for the resulting uncertainty in threshold position measurement to be in the same range as the other uncertainties. This way the influence of $\sigma_{\Delta \alpha }$ will not dominantly influence the overall uncertainty. The influence of $\sigma_{\Delta \alpha }$ is demonstrated in Figure 27.

### 5.5. Illumination Source Power

This section investigates the influence of variation in the illumination source power. Variations in the optical power of the laser could result from supply current fluctuations, as well as temperature changes. Because the testbed photo diode was used to measure the illumination source power, the determined uncertainty also includes the influence of the photo diode and the oscilloscope.

To gather measurement data for the power variation, the laser on the test-bed was configured to aim directly at the photo diode. Over the measurement time of 15 min, the drift of the signal was less than 0.125%. Its noise is normally distributed with a standard deviation of $\sigma_{P0}=0.26\%$. The simulation results for varying illumination source power are shown in Figure 28.

### 5.6. Edge Transition Threshold Detection

In this section, the influence of the threshold detection scheme is investigated. The photo diode picks up the transition between CFRP and chrome paint on the surface of the flywheel rotor, resulting in an analog edge transition signal. We want to find the center of the transition by sampling the signal and calculating the time the threshold was reached. The photo diode rise time is low compared to the transition function between materials, and can therefore be neglected [11].

For the uncertainty evaluation of the threshold detection scheme, we are interested in the time difference between the calculated and the true threshold position. The sampling rate of a typical microcontroller’s analog-to-digital converter (ADC) is too slow to sample the edge transition in the required time resolution of 3.3 ns, but we can sample the signal at a lower sample rate and interpolate between the data points. Calculating the threshold from the interpolated signal introduces an error in the threshold position. A simulation of sampling the edge transition at 5 MS/s (typical sampling rate of a microcontroller) with an appropriate anti-aliasing filter is shown in Figure 29. The sampled signal is subsequently interpolated to find the position when the threshold of 50% is reached. The linear interpolation is easy to implement and the sinc interpolation is an example of a more resource-intensive method.

The error introduced by different interpolation schemes is the time difference between the original signal and the interpolated signal at the threshold value. It is shown in Figure 30. The shift parameter represents the timing misalignment between the real signal and the sample time which can vary up to the sample period $T_{s}=1/f_{s}$ where $f_{s}$ is the sample rate. The shift is unknown and varying over time because the signal originates from the rotating mass of the flywheel. Slight variations in rotation speed influence the shift. If we assume the shift is uniformly distributed between 0 s and $T_{s}$, the error follows an arcsin distribution shown in Figure 31.

## 6. Conclusions and Summary

In this section, all investigated influences are discussed and compared against each other. Table 1 lists the uncertainty of each source parameter and the calculated threshold position uncertainties resulting from the individual simulations. For the paint edge noise, both examined variants are listed, and for each a separate total is calculated. The sources are assumed to be statistically independent from each other, and the total uncertainty is therefore calculated as
(6)$$\sigma_{\Delta ,Thresh,total,v}=\sqrt{\sum_{n}^{}\sigma_{\Delta ,Thresh,n}^{2}}$$
where *v* is either Variant A or B and $\sigma_{\Delta ,Thresh,n}$ are the uncertainties of the threshold position for each influence source *n*.

When we compare the listed uncertainties, we can loosely group them into categories: negligible, average, and dominant. Because of the squared weighting, the influence of the laser power noise, as well as the ADC threshold detection noise are negligible. In the average category, we find the paint reflectivity noise and the axial height noise (after correction of the systematic height difference). The sensor angle noise was deliberately chosen to also fall into the average category when the required parameter limit requirement is met.

The dominant influence on the total threshold position uncertainty is the paint edge noise, in either Variant A or B. Variant A—the paint edge produced by masking and spray painting—results in an uncertainty more than three times larger than the next largest entry. The largest reduction in total uncertainty can be achieved by focusing on the paint edge quality, as demonstrated by Variant B—the reworked paint edge. For a further reduction in the total uncertainty, improvements in all of the average sources of influence are required in addition to the paint edge. This will prove to be impractical, as discussed in the next paragraph.

The total threshold position uncertainties are $\sigma_{\Delta ,Thresh,total,A}=20.4\mu m$ for Variant A and $\sigma_{\Delta ,Thresh,total,B}=12\mu m$ for Variant B. Rearranging Equations (Equation 4) and (Equation 5) for $u_{min}$ and calculating the percentage with respect to du=121μm results in 6.1% strain accuracy for Variant A and 3.6% strain accuracy for Variant B, respectively. When we compare the total threshold position uncertainties to the initial requirement of 3.3 μm (1% strain accuracy), it is obvious that neither Variant A nor B can meet this criterion without averaging over multiple measurements. To fulfill the requirement and reduce the uncertainty, we have to average over multiple measurements via
(7)$$N_{v}={\left(\frac{\sigma_{\Delta ,Thresh,total,v}}{3.3\mu m}\right)}^{2}$$
where $N_{v}$ is the number of measurements to average, and *v* is either Variant A or B.

In consequence, $N_{A}=39$ measurements are required for Variant A and $N_{B}=14$ for Variant B. At the nominal speed of 500 Hz, we can expect a measurement rate of 80 ms and 28 ms, respectively. Both of those measurement rates are adequate for monitoring the changes in the CFRP strain. Since averaging over the measurement time is necessary anyway, we can increase the number of lobes to trade measurement time for a better angular resolution.

In summary, a ray tracing simulation framework for an optoelectronic strain measurement system was developed. The framework allows the simulation of the influence an uncertain model parameter has on the strain measurement. Based on measurement results or calculations, influences for the common parameters in the OESM setup were assigned uncertainty values. The resulting uncertainties on the threshold position measurement were calculated and discussed.

## Figures and Tables

**Figure 1 sensors-21-08393-f001:**
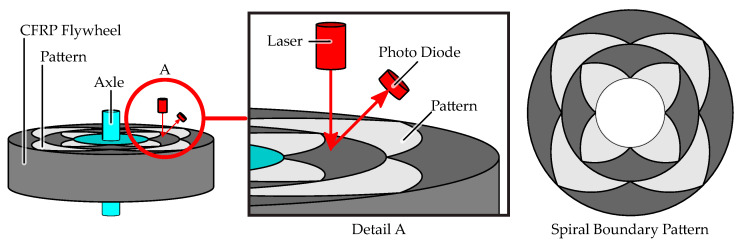
Basic operation principle of optoelectronic strain measurement (OESM) on a flywheel. A special pattern consisting of pieces of Archimedean spirals is applied to the rotor. The laser illuminates a spot on the rotor surface, and the photo diode is used to measure the reflected light.

**Figure 2 sensors-21-08393-f002:**
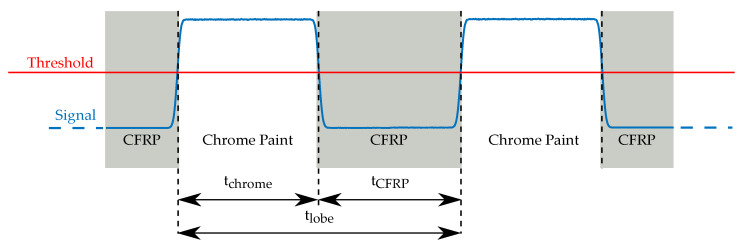
Electrical output signal from the OESM system. The transition from CFRP to chrome paint is called the edge transition function and is not an ideal step function. A threshold is applied to determine the time of the transition.

**Figure 3 sensors-21-08393-f003:**
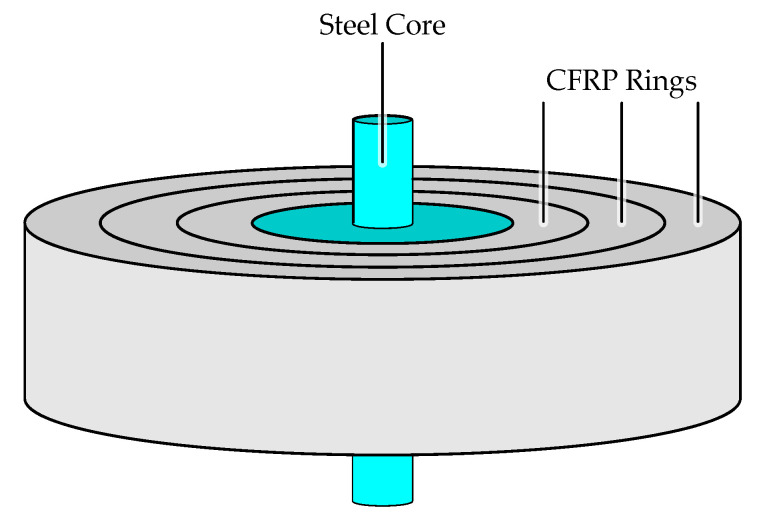
Construction of the flywheel. Three CFRP rings are press-fitted onto a steel core.

**Figure 4 sensors-21-08393-f004:**
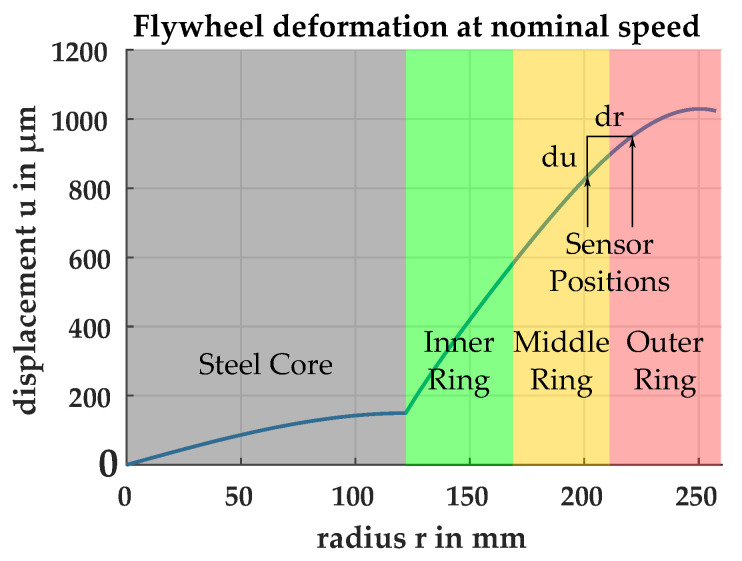
Flywheel ring layout and deformation at nominal speed. The relative deformation decreases with radius. The interface between the middle and outer ring was chosen for calculating the accuracy requirements because it is the point of interest with the lowest relative deformation and therefore the worst case requirement.

**Figure 5 sensors-21-08393-f005:**
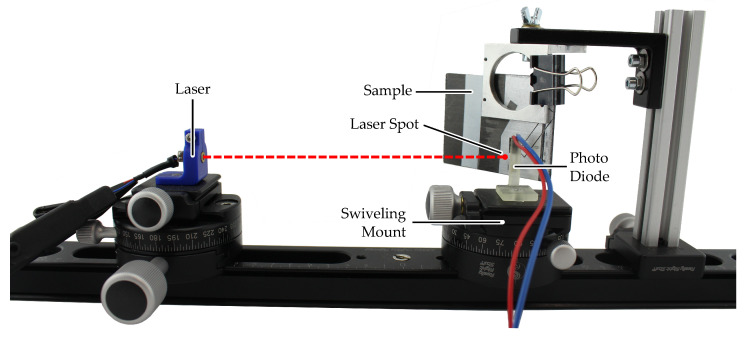
Measurement setup for directional reflectance measurement of painted surfaces. The laser illumination source is mounted on the left and adjusted so that the laser spot is focused on the material sample surface. The photo diode can be rotated around the center of the mount, keeping the distance between sample surface and photo diode constant.

**Figure 6 sensors-21-08393-f006:**
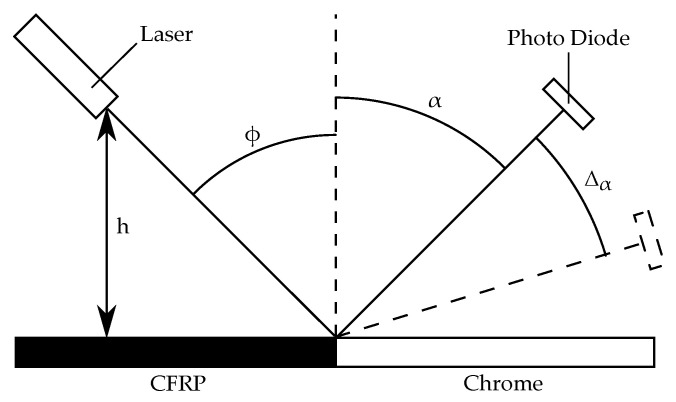
Schematic of the measurement setup for directional reflectivity measurement of painted surfaces. The ray tracing simulation setup also follows this arrangement.

**Figure 7 sensors-21-08393-f007:**
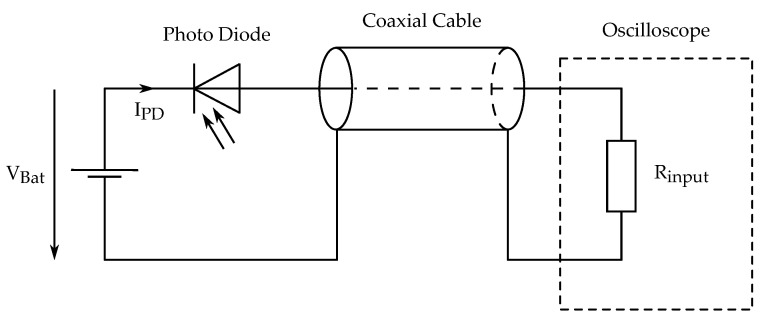
Schematic of the light detector.

**Figure 8 sensors-21-08393-f008:**
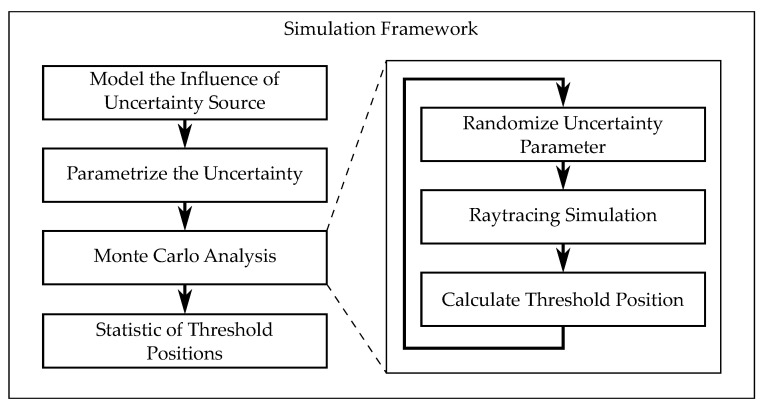
Simulation framework. Steps done for each source of uncertainty.

**Figure 9 sensors-21-08393-f009:**
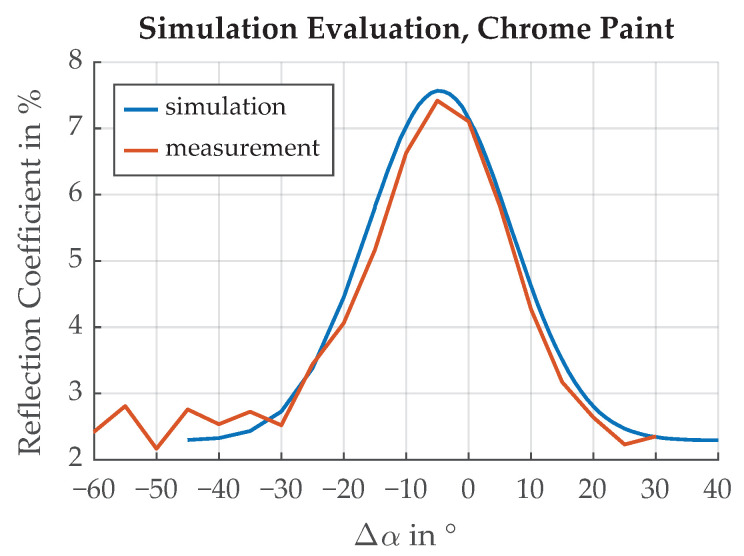
Comparison of the measured surface reflection with the simulated results for chrome paint. Incidence angle $\phi =45{}^{\circ }$, emergent angle $\alpha =45{}^{\circ }$.

**Figure 10 sensors-21-08393-f010:**
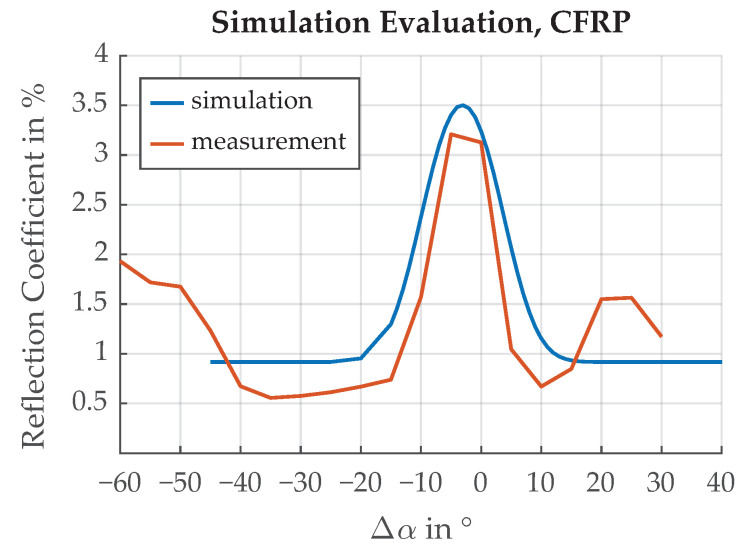
Comparison of the measured surface reflection with the simulated results for CFRP. Incidence angle $\phi =45{}^{\circ }$, emergent angle $\alpha =45{}^{\circ }$.

**Figure 11 sensors-21-08393-f011:**
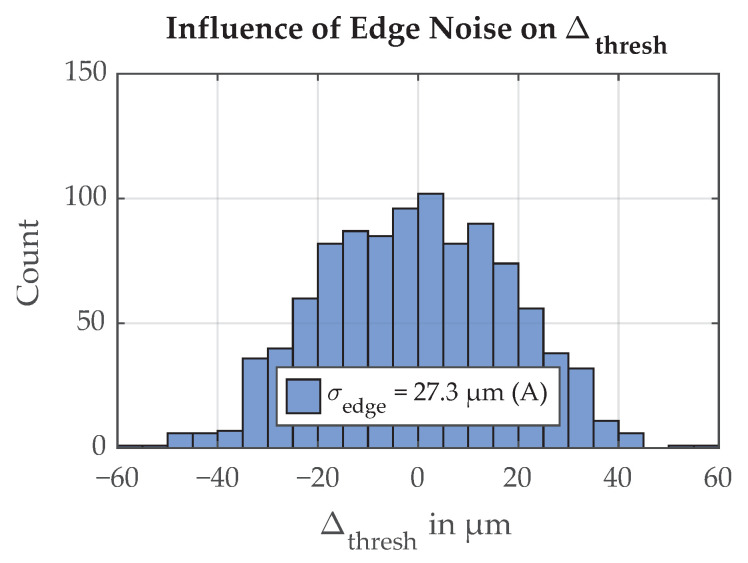
Histogram of simulated deviations in threshold position, edge noise Variant A, 1000 simulation repetitions. The resulting uncertainty is $\sigma_{\Delta ,thresh}=18.9\mu m$.

**Figure 12 sensors-21-08393-f012:**
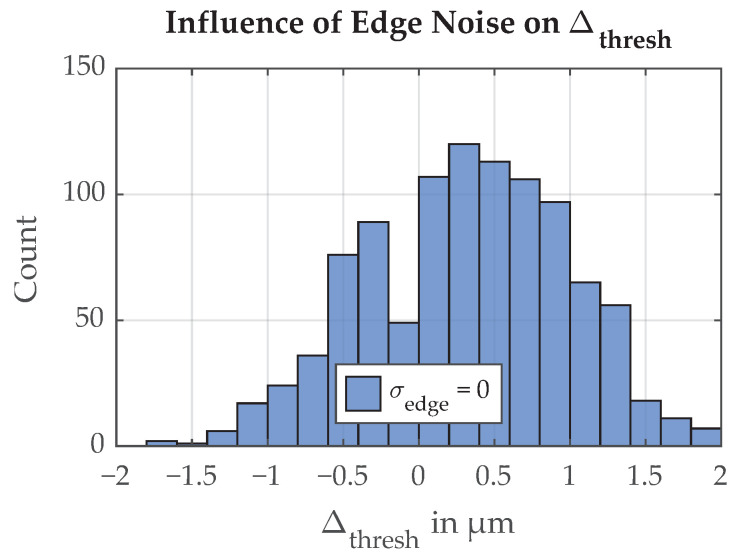
Histogram of simulated deviations in threshold position $\Delta_{thresh}$, simulation noise, 1000 simulation repetitions. The simulation noise comes from the discretization of the ray tracing. The resulting uncertainty is $\sigma_{\Delta ,thresh}=0.6\mu m$.

**Figure 13 sensors-21-08393-f013:**
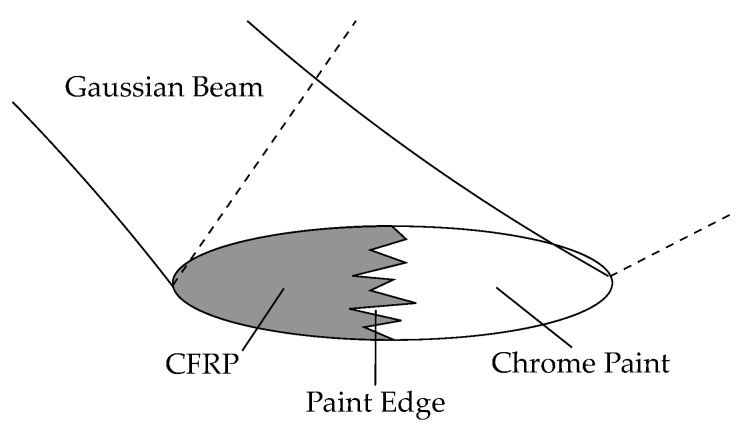
Model of the paint edge under the illuminated spot. The paint edge is not a straight line but rough.

**Figure 14 sensors-21-08393-f014:**
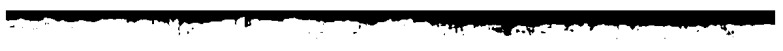
Chrome paint edge sample under the microscope, Variant A, binarized. One pixel equals 7.7μm.

**Figure 15 sensors-21-08393-f015:**
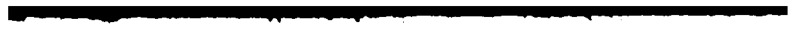
Chrome paint edge sample under the microscope, Variant B, binarized. One pixel equals 7.7μm.

**Figure 16 sensors-21-08393-f016:**
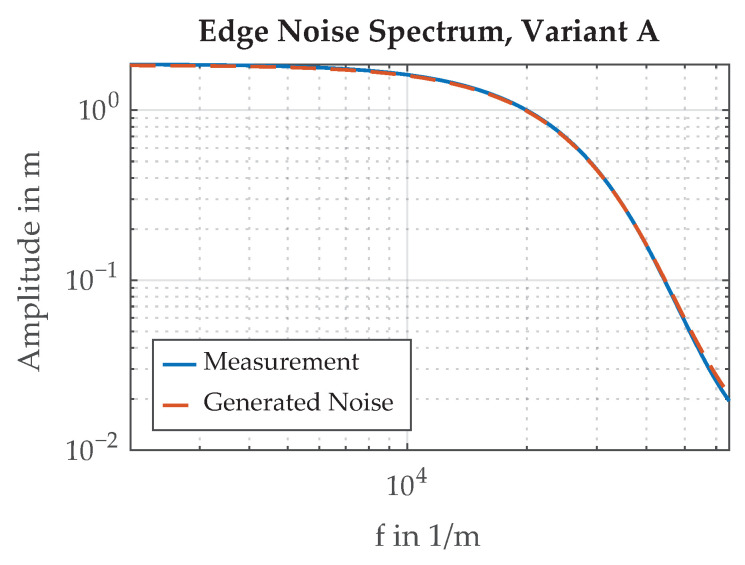
Frequency spectrum of the paint edge, Variant A, normalized to σ=1. The generated noise has a similar spectrum to the measured edge noise.

**Figure 17 sensors-21-08393-f017:**
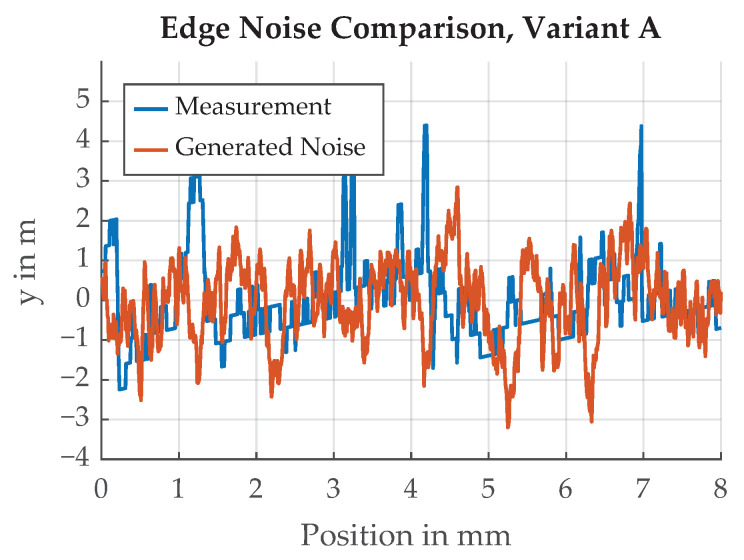
Qualitative comparison of measured and generated paint edge, Variant A, normalized to σ=1.

**Figure 18 sensors-21-08393-f018:**
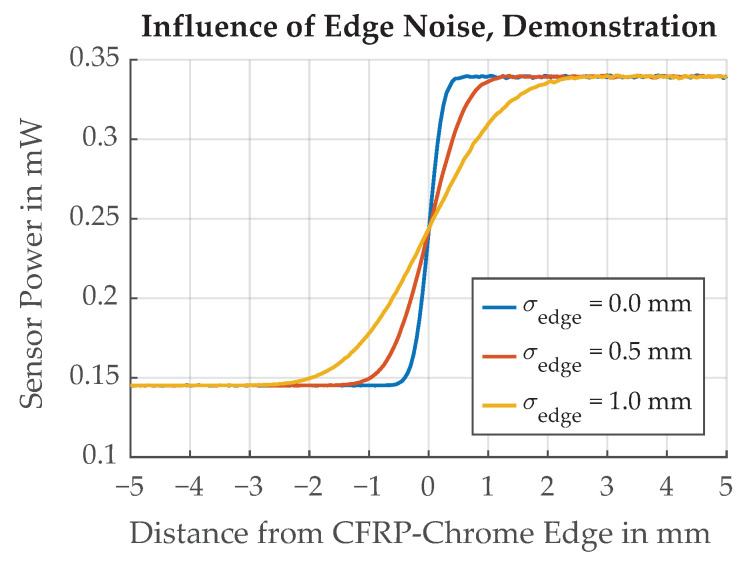
Simulation results for variation of $\sigma_{edge}$ to demonstrate the influence of edge noise on the transition function.

**Figure 19 sensors-21-08393-f019:**
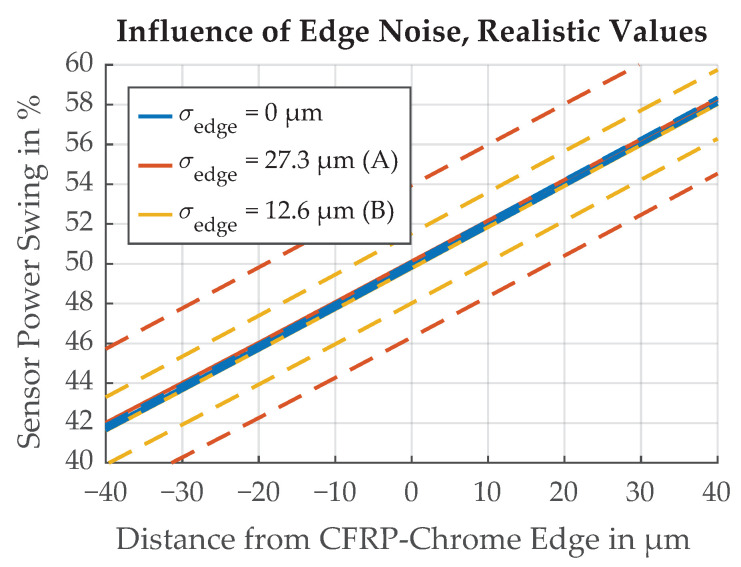
Simulation results of the influence on threshold position for variation of $\sigma_{edge}$ for realistic values from Variants A and B. Dotted lines represent ±σ.

**Figure 20 sensors-21-08393-f020:**
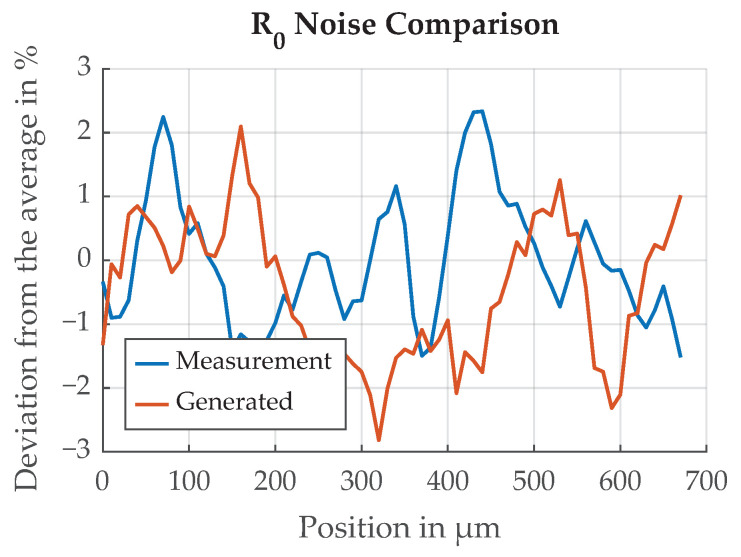
Qualitative comparison of the measured deviation of reflectivity $R_{0}$ and a generated random signal with the same standard deviation and similar spectrum.

**Figure 21 sensors-21-08393-f021:**
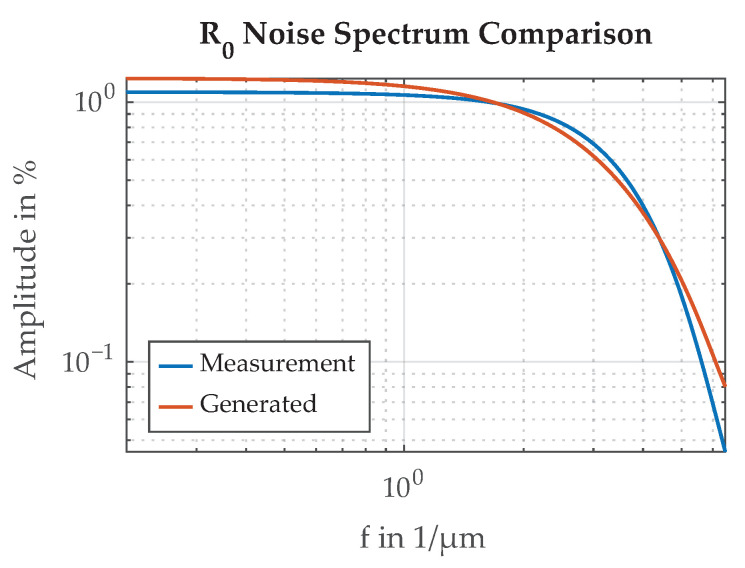
Spectrum of the measured deviation of reflectivity $R_{0}$ from the mean (normalized to 100%) and a generated random signal with the same standard deviation and similar spectrum.

**Figure 22 sensors-21-08393-f022:**
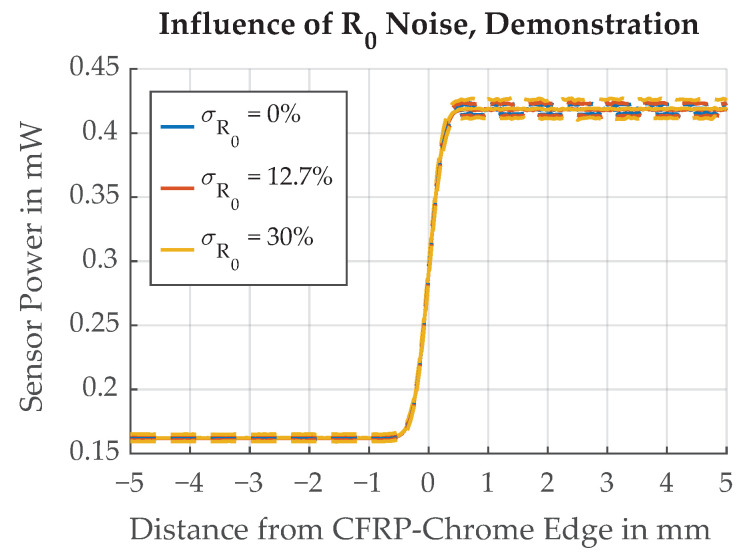
Demonstration of the influence of different standard deviations of the reflectivity $\sigma_{R0}$ on the edge transition function. Dotted lines represent ±σ.

**Figure 23 sensors-21-08393-f023:**
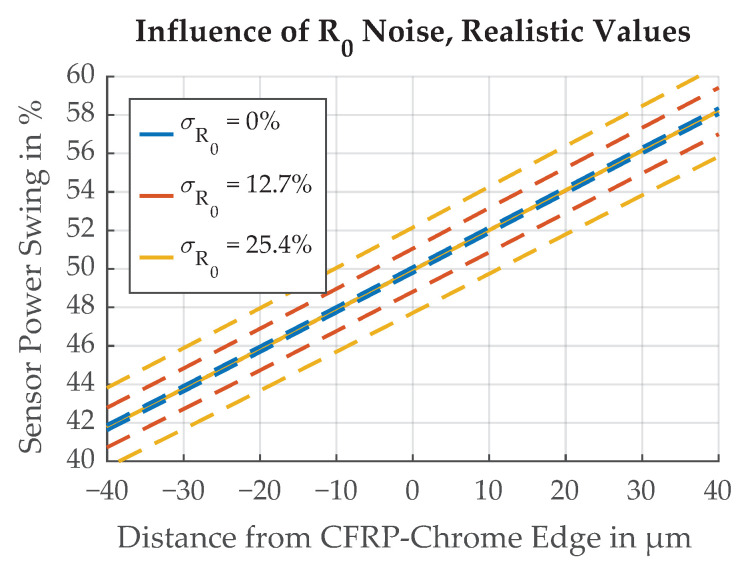
Influence of different standard deviations of the randomized reflectivity $\sigma_{R0}$ on the threshold position $\Delta_{thresh}$. Dotted lines represent ±σ.

**Figure 24 sensors-21-08393-f024:**
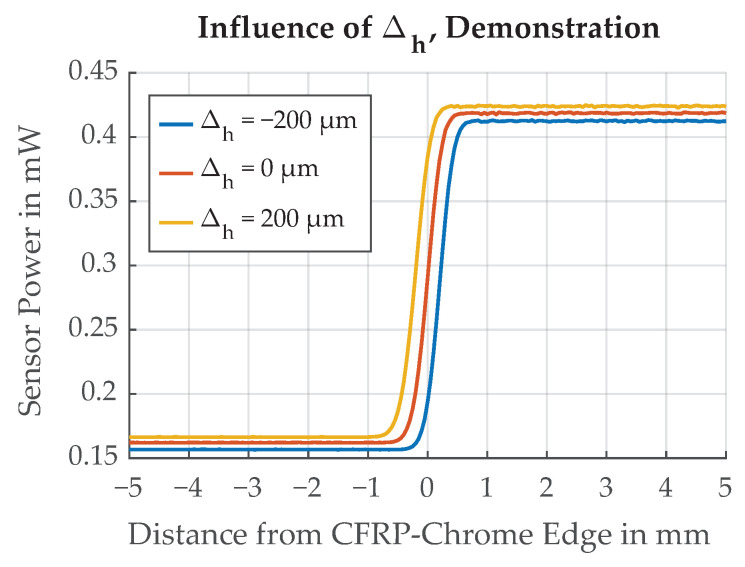
Demonstration of the effect of different values of $\Delta_{h}$. The influence of $\Delta_{h}$ is comparatively large and has to be corrected.

**Figure 25 sensors-21-08393-f025:**
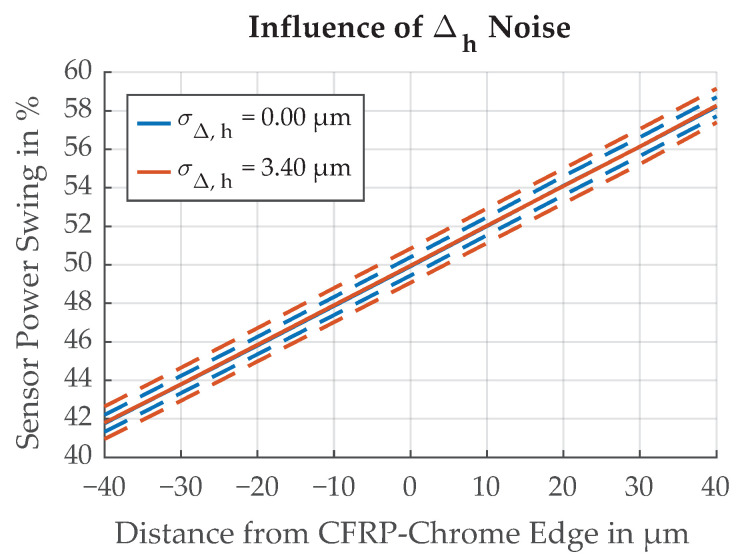
Influence of noise in $\Delta_{h}$ on the threshold position $\Delta_{thresh}$. Dotted lines represent ±σ.

**Figure 26 sensors-21-08393-f026:**
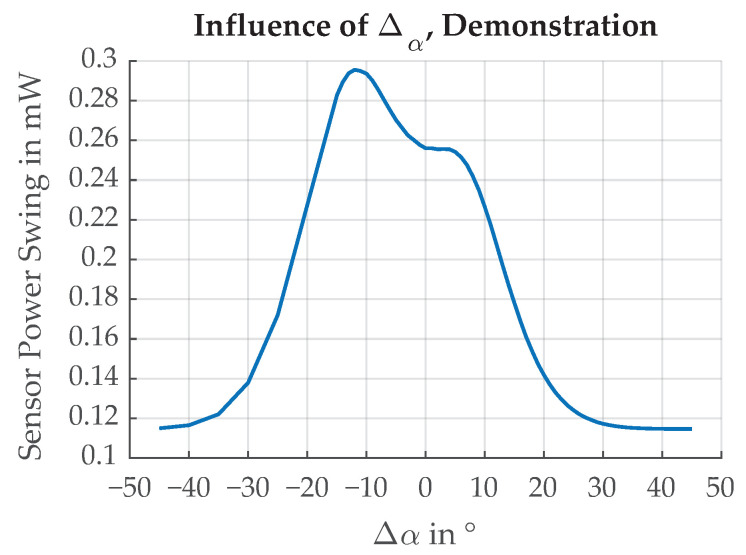
Sensor power swing (maximum value–minimum value) over different emergent angle shifts $\Delta_{\alpha }$. The maximum swing is at $\Delta_{\alpha }$ = −12${}^{\circ }$.

**Figure 27 sensors-21-08393-f027:**
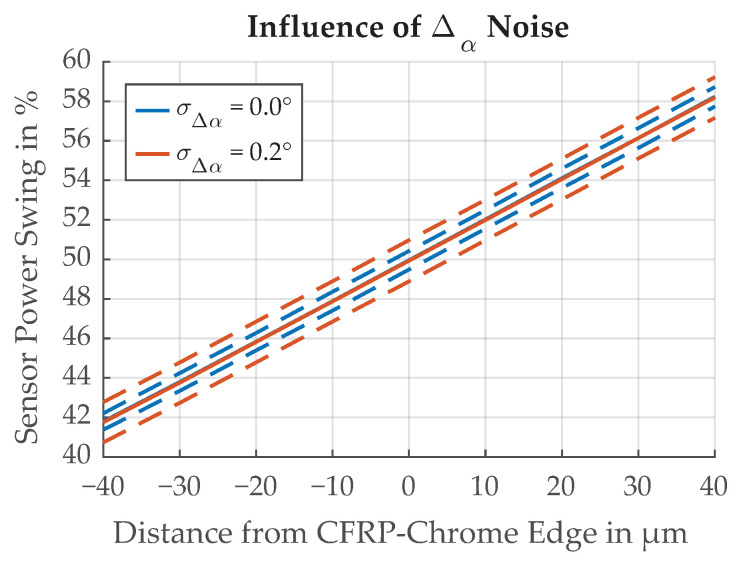
Influence of $\Delta_{\alpha }$ uncertainty on the threshold position $\Delta_{thresh}$. Dotted lines represent ±σ.

**Figure 28 sensors-21-08393-f028:**
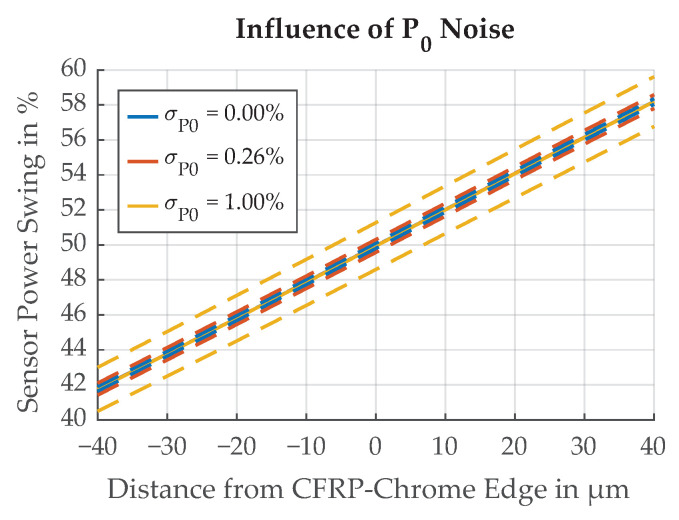
Influence of noise in the illumination source power $P_{0}$ on the threshold position $\Delta_{thresh}$. Dotted lines represent ±σ.

**Figure 29 sensors-21-08393-f029:**
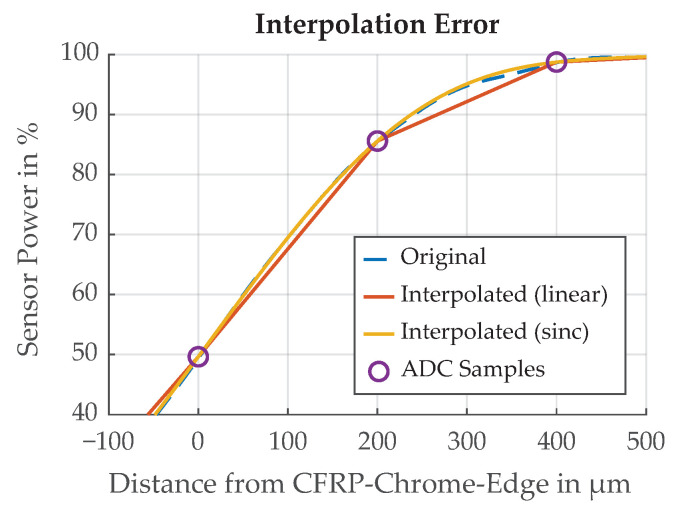
A simulation of sampling the edge transition function at 5 MSps and subsequent interpolation.

**Figure 30 sensors-21-08393-f030:**
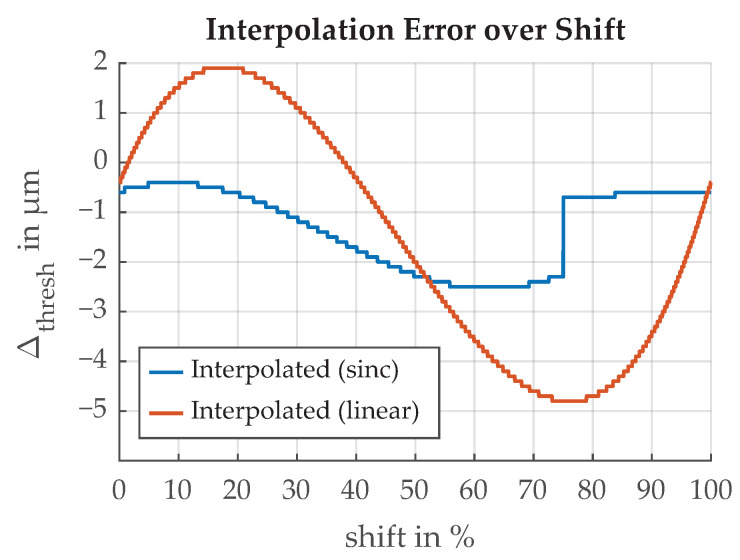
Error in threshold time introduced by sampling the edge transition function at 5 MS/s and subsequent interpolation. The shift parameter represents the location of the original analog signal between two sample points.

**Figure 31 sensors-21-08393-f031:**
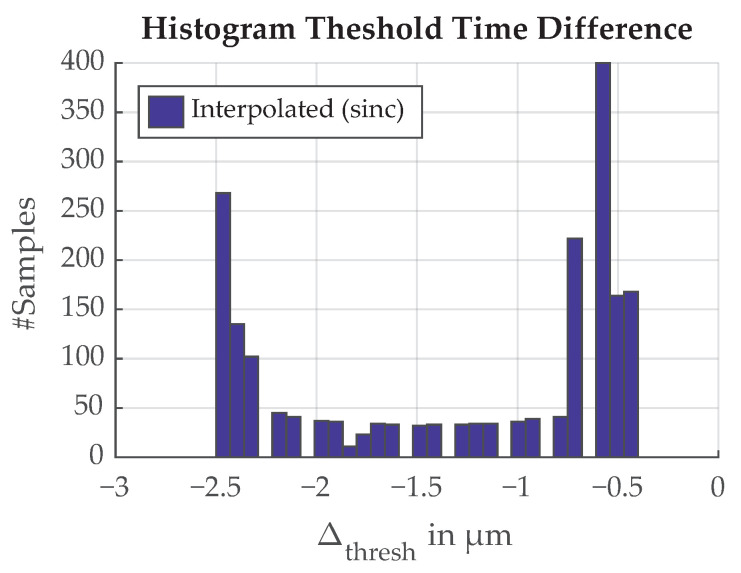
Distribution of the error introduced by sampling the edge transition function 5 MS/s and subsequent interpolation.

**Table 1 sensors-21-08393-t001:** Overview of all investigated sources of uncertainty converted to distance between detected and true threshold position and their relative contributions to the total variance $\sigma_{\Delta ,Thresh,total,A}^{2}$ for Variant A.

Source Parameter	Source Symbol	Source Value	Threshold Position Uncertainty	Relative Contribution to Total (A)
			$\sigma_{\Delta ,Thresh,n}$	
			µm	
Paint Edge Noise, Variant A	$\sigma_{edge,A}$	27.3 μ m	18.6	82.8%
Paint Edge Noise, Variant B	$\sigma_{edge,B}$	12.6 μ m	8.5	
Paint Reflectivity Noise	$\sigma_{R0}$	12.7%	5.4	7.0%
Axial Height Noise	$\sigma_{\Delta h}$	3.4 μ m	4.1	4.0%
Sensor Angle Noise	$\sigma_{\Delta \alpha }$	0.2 ${}^{\circ }$	4.7	5.3%
Laser Power Noise	$\sigma_{P0}$	0.26%	1.7	0.7%
ADC Threshold Detection Noise			0.8	0.2%
**Total, Variant A**			σΔ,Thresh,total,A = 20.4	
**Total, Variant B**			σΔ,Thresh,total,B = 12.0

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
