# Peer review of "Uncertainty Analysis of an Optoelectronic Strain Measurement System for Flywheel Rotors"

_sensors, 2021, doi:10.3390/s21248393_

Round 1

Reviewer 1 Report

Title: Uncertainty Analysis of an Optoelectronic Strain Measurement System for Flywheel Rotors

Overview and general recommendation:

In this manuscript, the factors affecting the strain of flywheel rotors are analyzed from the direction of strain uncertainty. Monte Carlo method is used to analyze the uncertainty of each key component. The comprehensive analysis of all kinds has certain reference significance for the continuous optimization of optoelectronic strain measurement system for flywheel rotors reliability and accuracy. Finally, the influence of the uncertainty of each component on the system is obtained.

On the other hand, I found the paper to be overall well written and much of it to be well described. I felt confident that the authors performed careful experimental analysis and simulation. Additionally, I found the description of the paper missing analysis of several uncertain components. Therefore, I recommend that a minor revision is warranted. I explain my concerns in more detail below. I ask that the authors specifically address each of my comments in their response.

2.1. Major comments:

The components of the flywheel system will produce rigid body vibrations in and out of the plane at different speeds. However, the manuscript does not seem to include the analysis and discussion of this part of the impact. I presume the analysis of the vibration part is very necessary for the credibility of the system.

Section 3.1. The ray tracing simulation mentioned in the manuscript also has no simulation process and no mention of the software tools used in the simulation. The description was inadequate or completely missing. I have very little confidence in one important analysis, and came away with too many questions.

Section 5.3. This section discussed the uncertainty caused by spot size, and assuming that the light spot size and divergence angle as a constant value. There is no problem with the above assumption. However, the light-emitting characteristics of semiconductor lasers determine that the collimation of laser is usually elliptic. If the illumination spot is not nearly circular, it will affect the test results. It ends up being such a critical step in your processing. The author should show the really characteristics of the illumination spot shape.

Section 5.6. The response time of the photodiode is also a key parameter of the test system. It should be explained whether the rise time of the photodiode increases the uncertainty of the system. Otherwise, just concisely explain why there is no mention of photodiode rise time. Or authors think that this parameter does not affect the test results.

2.2. Minor comments:

  1. Page 6: line 163, Page 7, line 176: Notice the order of the two pictures;

Page 6: lines 148-149: The measurement values mentioned here can be added to Figure 6 to improve the reliability of the simulation results;

Page 10: figure 17 does not seem to be referenced;

Page 10: lines 224-226: Consider quoting from previous articles;

Page 10: lines 228-231; Page 11, lines 236-240; Notice the order of Figure 20, 21, 22;

Page 15: Figure 30: Please list the remarks of the two sets of data

Reviewer 2 Report

This is a nice presented work. It aims at measurnig strain in a fast spinning carbon fiber flywheel rotor based on an optoelectronic system. The measurement uncertainty is handled and analyzed. It is of interest to the society. 

The MCS is used for uncertainty tracking. However, it may be less efficient due to large number of samples needed. There is plenty of literatures on the rotor system modelling with random or non-random uncertainties. The authours can read them and take any reference.

Reviewer 3 Report

The research focuses on analysing the uncertainty of measuring strain in flywheel rotors  using optoelectronic measurement system. Overall, the authors just show what they have done, but there are not strong connections between each section. There are not scientific supports in many statements and research decisions. The research results do not support the objectives of the research Below are comments in details:

  1. The research results need to be presented in the abstract. Currently, it generally states the research results which can be found in many researches in the same field.
  2. The introduction cannot highlight reasons for conducting the research. The introduction does not briefly and specifically give the methodology, achievement results of the research.
  3. There are no connections between each section in the manuscript, which make the manuscript illogical and difficult to understand the roles of each research component to the outcome of the research.
  4. There are no scientific supports for many statements. For example, why have the authors selected the final outer radius of 260 mm (the authors cited references, but the outer radii in references are not 260 mm)? How have the authors done to have the relative deformation du=121 um for the distance dr = 20 mm between two sensors shown in page 4? What are the reasons that the authors required an accuracy of at least 1%? Why do the authors require the 3․σ uncertainty of the measurement is less than ± dD,Thresh/2?
  5. As a reader, I do not understand the meaning of section 3 (simulation framework) in this research until I read section 4. However, there are many unclear points in section 3. For example, how the simulation model has been built (which software have the authors used, which algorithm have the authors used to collect between input and output values, what is the accuracy of the simulation model, etc.)
  6. The research is on the uncertainty of measuring strain on flywheel using optoelectronic measurement system. However, authors have not shown how to calculate/measure the uncertainty.
  7. It is not clear the relationship between measuring reflection on a static surface with the accuracy in measuring strain in a rotating flywheel. Therefore, the meaning of section 4 (measurement setup) in this research is not clear.
  8. How parameters mentioned in section 5 (Uncertainty Analysis) affect to the accuracy in measuring strain using optoelectronic measurement system has not been reported. For example, how much does the paint edge influence the strain measurement?
  9. 90o section (page 5) should be a quarter.

Round 2

Reviewer 3 Report

The revised manuscript and the response to the reviewers are quite satisfactory from my perspective. However, the response of the authors to the concern relating to the connections between sections in the manuscript should be improved. Currently, the authors listed title of each section in the last paragraph of section one. This listing style is weird for a research article, which is not review article.
